# Mind the gap between recommendation and implementation—principles and lessons in the aftermath of incident investigations: a semi-quantitative and qualitative study of factors leading to the successful implementation of recommendations

Jonas Wrigstad,[1,2] Johan Bergström,[3,4] Pelle Gustafson[2]

For numbered affiliations see end of article.

**Correspondence to**
Dr Jonas Wrigstad;
jonas.wrigstad@med.lu.se

## ABSTRACT

**Objectives:** Using the findings of incident investigations to improve patient safety management is well-established and mandatory under Swedish law. This study seeks to identify the mechanisms behind successful implementation of the recommendations of incident investigations.

**Setting:** This study was based in a university hospital in southern Sweden.

**Participants:** A sample of 55 incident investigations from 2008 to 2010 were selected from the hospital's incident reporting system by staff in the office of the chief medical officer. These investigations were initiated by 23 different commissioning bodies and contained 289 separate recommendations. We used a three-stage method: content analysis to code the recommendations, semi-structured interviews with the commissioning bodies focusing on which recommendations had been implemented and why, and data analysis of the coded recommendations together with data from the interviews.

**Results:** We found that a clear majority (70%) of the recommendations presented to the commissioning bodies were targeted at the micro-level of the organisation. In nearly half (45%) of all recommendations, actions had been taken and a clear majority (73%) of these were at the micro-level. Changes in the management positions of the commissioning bodies meant that very little further action was taken. Other actions, independent of incident investigations, were often taken within the organisation.

**Conclusions:** We conclude that two principles ('close in space' and 'close in time') seem to be important for bridging the gap between recommendation and implementation. The micro-level focus was expected because of the method of investigation used. Adverse events trigger organisational action independently of incident investigations.

### Strengths and limitations of this study

- The results presented in this study show the strength of using a design that combines content analysis with interviews to thereby provide deeper understanding of the different aspects of the data.
- The semi-structured nature of the interviews seemed to make the respondents willing to elaborate and reflect freely on both questions and follow-up questions, which resulted in a substantial amount of qualitative data.
- The coding scheme used in the content analysis and the categories used in the data analysis could have possibly resulted in a limited perspective of a more complex reality.

## INTRODUCTION

When adverse events (AEs) occur in complex socio-technical healthcare systems, it is difficult—if not impossible—to identify the underlying causal factors. The importance of using past events to promote organisational learning is obvious but hard to institutionalise in practice.[1] Nevertheless, incident investigations have for decades been routine and regarded as important tools in safety management, primarily to prevent similar events occurring again by promoting recommendations for ensuring continuous improvement.

Different organisations use different methods to conduct incident investigations, with the majority, including healthcare, having adopted an underlying accident model in which recommendations made assume the system has a stable causal

structure.[2][3] This stable causal structure implies that the recommendations are derived by identifying the root cause with no need to relate the specific recommendations to the damaged system as a whole.[2] Johnson argues that understanding of how certain recommendations are formulated is generally weak.[4] It has been shown that investigators spend a surprisingly short amount of time providing recommendations in comparison to other parts of the process.[5] Furthermore, the factors governing successful implementation of recommendations have so far received limited attention in the literature.[6][7]

The aim of this study was to start filling this knowledge gap by analysing the mechanisms behind the successful implementation of recommendations formulated in investigations of incidents in Swedish healthcare. The approach follows Hollnagel's advice to search for the positive rather than the negative aspects of safety.[8]

## BACKGROUND

The Swedish healthcare system's regulatory authority at the time of the study, the National Board of Health and Welfare (NBoHaW), has issued regulations governing the responsibilities of the different healthcare providers, for example, when using an incident reporting system and carrying out incident investigations. Swedish law states that the responsibility for patient safety improvement lies with the separate healthcare providers.[9][10] The law also states that if an AE has resulted, or could have resulted, in a serious incident, this should be reported to the regulatory authority for separate investigation. This investigation, the so-called Lex Maria (LM) investigation, is independent of the incident investigation conducted by the healthcare provider. The chief medical officer (CMO) of an organisation generally decides whether or not to report an AE to the NBoHaW, although the CMO has neither formal legal authority nor responsibility for the safety level of the organisation.

A commissioning body (CB) initiates and sets the terms of reference for the incident investigation, and is ultimately responsible for follow-up of the report recommendations.

The analysis team, set up by the CB, consists of at least one healthcare professional trained in investigating AEs in the Swedish healthcare system. Since 2005, methodological support for conducting investigations has been provided by the Swedish Association of Local Authorities and Regions (SALAR) and supported by the NBoHaW.[5] In Swedish healthcare, completed incident investigation reports are, after de-identification, made publicly available, as are LM investigations conducted by the NBoHaW.

## METHODS

We used a three-stage method. First, we carried out content analysis to code the recommendations in a sample of 55 incident investigations of AEs in a Swedish university hospital. We then conducted semi-structured interviews with CBs focusing on which recommendations had been implemented and why. Finally, we performed data analysis using the coded recommendations together with the interview data, to identify specific mechanisms contributing to successful implementation of recommendations. Due to the semi-quantitative nature of the study, we carried out no further statistical analyses.

## Content analysis

The first step was to sample a limited number of completed incident investigations. In collaboration with the CMO at a Swedish university hospital, data on registered AEs which resulted in incident investigations were collected from the hospital incident reporting system. The CMO was asked to determine for which years after 2005 (when the methodological support manual by SALAR was published) the hospital had sufficient qualified incident investigator staff working within the organisation and familiar with the methodology. Second, at least 1 year should have elapsed after completion of the incident investigation to allow for the implementation of recommendations. Third, the selection of investigations should be linked to incidents in which the department of anaesthesia and intensive care was involved as the main author is an anaesthesiologist, ensuring (1) a comprehensive data set through contacts with important actors, as well as (2) full understanding of the cases and investigations, regardless of complexity. This resulted in the selection of 55 separate incident investigations from January 2008 to December 2010, initiated by 23 different CBs. We also identified the staff position initiating the incident investigation, as this was the same position to which the recommendations would be presented upon completion. Thus, continuity in management was of interest, not individuals.

The completed incident investigations were linked to existing additional investigations, for example, LM investigations, using the hospital incident reporting system.

All incident investigation reports and recommendations were numbered as they were received from the office of the CMO. Data from the reports were coded according to the CB at the time of investigation, the ward from which the analysis was commissioned, the time spent by the team conducting the investigation, the number of team members, the number of suggested recommendations, and whether or not the findings of the investigation were reported by the hospital to the NBoHaW (as an LM investigation).

Rasmussen and Svedung[11] have shifted the focus to include 'what' causal factors are identified in the aftermath of AEs, and 'where' in the organisational hierarchy the identified causal factors are. We therefore coded the reports according to the hierarchical level of the target of the recommendations using a micro-meso-macro perspective.[12] This was done in order to identify potential correlations between hierarchical level and the likelihood of the recommendation being implemented. A micro-level recommendation could be implemented by

 Wrigstad J, Bergström J, Gustafson P. *BMJ Open* 2014;**4**:e005326. doi:10.1136/bmjopen-2014-005326

the CB entirely within the same department without major constraints, for example, as regards local procedures, technical skills or staff issues. With a meso-level recommendation, the CB had to collaborate with a stakeholder outside the department but within the hospital, for example, another department or the hospital management. With a macro-level recommendation, the boundaries of the hospital had to be crossed, for example, authorities, politicians or pharmaceutical companies had to be contacted.

From the written reports it was not possible to determine to what extent the different recommendations had been implemented or not. These data were added to the coding scheme following the interviews.

### Semi-structured interviews

The second part of this study consisted of interviews with the different CBs at the hospital, to gain deeper insight into which recommendations had been implemented and why. The interviews were semi-structured as they focused on specific reports, but with the possibility for the respondents to reflect freely on the questions asked.

All of the CBs received written information before the interview about the background and aims of the project, as well as the main questions forming the basis of the interview. All respondents were de-identified and given a random number. Twenty-two of 23 CBs (or their successors) provided written consent to being interviewed. This made it possible to ask questions of interest about 50 of 55 incident investigation reports, with a total of 254 coded recommendations. Four of the 22 CBs delegated the interview to either an assistant director (2/4) or the head advisor in patient safety (2/4). The interviews were all carried out between April and September 2012 by the first author (JW) at a place suggested by the respondent. Twenty of the interviews were audio recorded. In two of the interviews the respondents did not agree to audio recording and so extensive notes were taken instead. All quotations presented here have been translated from Swedish to English by the first author and are all tagged with the number of the coded respondent.

All interviews included a minimum of three questions (see below). Subsequent questions were asked depending on the answers given by the respondents.
1. Have you taken part in this incident investigation report and given attention to the recommendations *before* this study?
2. Which recommendations from the incident report have been *implemented* in the organisation?
3. Have, to your knowledge, any *alternative* actions been taken within the organisation because of the incident investigation report that were not presented as recommendations?

During the period studied, the hospital had a system where in one part of the hospital the CB was nearly always the CMO, while in the other part the CB was the

clinical head of department. In addition, a CMO in a Swedish hospital cannot also be clinical head of a department at the same time. Therefore, for investigation reports where the CB was the CMO, interviews were also conducted with the clinical heads of the departments involved in order to gain deeper knowledge of how far the implementation of recommendations to the different departments had progressed.

### Data analysis

The analysis used the interview data to seek naturalistically generalised factors explaining the results of the content analysis.[13 14]

Before naturalistic generalisation, the coding scheme was extended to include the different answers as to whether action had been taken or not on specific recommendations. We used three categories in this study:
1. Actions have been taken and initiated/completed regarding the recommendation
2. Actions have not been taken regarding the recommendation
3. No knowledge if actions have been taken.

As many clinical heads of department would be interviewed about the same specific incident investigation, some of their answers might be assigned to conflicting categories. It was therefore important to follow up answers to category A with questions about how and when the particular recommendation had resulted in actions.

We analysed interview data in a search for generalised patterns: Why did the distribution between micro-level, meso-level and macro-level recommendations look the way it did? What was the connection with successful implementation, and why? What aspects of successful recommendation implementation were not captured in the content analysis, and why? Did, or did not, factors such as the position of the CB or the time spent by the analysis team, influence the likelihood of the suggested recommendations being implemented?

### RESULTS
#### Content analysis of incident investigation reports

Thirty-nine of the 55 AEs were subject to both an incident investigation by the hospital and to an LM investigation by the authorities, suggesting that the severity of the AE in most events had exceeded an official threshold. Implementations of recommendations from LM investigations were not analysed in this study.

The CBs of the 55 incident investigations were similarly distributed between CMO (n=29) and heads of department (n=26).

The average number of team members per investigation was 2.7, and the duration of an investigation varied from 12 to 150 man-hours, similar to the findings by Rollenhagen for typical investigations in patient safety.[5]

A total of 289 separate recommendations were identified in the 55 incident investigations, with five

Table 1 Distribution of the recommendations according to the three hierarchical levels and whether they were reported as having resulted in actions

| Hierarchical level | No. of recommendations for which actions have been taken | No. of recommendations for which actions have not been taken | No. of recommendations for which it is not known if actions have been taken | Total number |
|---|---|---|---|---|
| Macro | 1 | 6 | 0 | 7 |
| Meso | 30 | 26 | 16 | 72 |
| Micro | 82 | 53 | 40 | 175 |
| Total number | 113 | 85 | 56 | |

recommendations not coded due to uncertainty concerning the meaning of the investigators' findings. Thus 284 coded recommendations were included and questions about 254 of them were asked during the interviews. The distribution of these recommendations in the organisational hierarchy is shown in table 1.

In the following sections semi-quantitative and qualitative data, including the categories from the content analysis and quotations from interviews with CBs, will be presented in order to identify mechanisms important (or not) for the successful implementation of recommendations.

## Management continuity

The interviews revealed that the hospital, after commissioning the investigations, had replaced one CMO. This CMO was involved in 29 incident investigations, where one incident investigation could involve a number of department directors. The interviews also revealed that in 41 cases, regardless of the position of the CB, the clinical heads of department also had been replaced. When the question 'Have you taken part in this incident investigation report and given attention to the recommendations before this study?' was asked, the new CMO had taken part in 3/29 investigations, as did 6/41 of the new clinical heads of department. As one of the CBs noted:

One could have a system where the CMO is a bit more meticulous and does a follow-up of the incident investigations to see what happened. It could be more of a supervising position than it is today, but there is no time for that. That would probably be a part time job in itself or a substantially increased workload. (11)

Overall, the respondents were concerned about lack of knowledge regarding incident investigation reports completed before they assumed their current management position:

I have not informed myself about past events, but that illustrates two important things, according to myself, that we use the results from the incident investigations too scantly and there is not enough follow-up … But I think the most important matter is – these are historical cases and if one hasn't been clinically involved it's a problem with commitment – that there is a follow-up on the recommendations so that something does happen. (2)

No, note that there isn't a single one of these incident investigations I've known about […] I've talked to my assistant director [a doctor] and to the member of staff responsible for the departments incident reporting [a nurse] to collect some information. (22)

Nowhere in the organisation did we find a proper system for recording what actions had been taken following the recommendations of the incident investigations. To varying degrees, the respondents had been able to find information on what actions had been taken. As shown in table 1, actions had been taken for 45% of recommendations, actions had not been taken for 33% of recommendations, and our respondents were unable to tell us whether or not actions had been taken for 22% of recommendations.

## The position of the CB

Whether the CB was a CMO or head of department did not seem to influence the process of implementation. When it was a CMO, actions had been taken in 55/128 (43%) of completed investigations, and when it was a head of department in 58/126 (46%):

…and when many departments are involved in the adverse event it doesn't work with just one director of department being the commissioning body … But it's also complicated to hand this over to the chief medical officer because it often has a tendency to come to nothing when many actors are involved. Who takes the responsibility? (11)

## Micro, meso or macro

As seen in table 1, in the cases where actions had been taken, the interviews showed that a majority were at the micro-level:

Yes, actions have been taken. We've written a new document about this procedure, that I have right in front of me, so that I can remember everything that has been done … and regarding that matter, we've put it on the checklist and the surgeon must ask before surgery whether procedures have been followed … This was a very easy and straightforward thing to solve, one could say. There was one thing that had gone wrong and we tried to fix it … and others weren't involved. (4)

## The event itself as a trigger for change

In 19 of 50 cases, the interviews showed that the AE had initiated organisational actions that were not presented as recommendations in the reports. It seemed that the incident investigations in these cases worked more as an incentive for change, but on the initiative of management rather than the analysis team:

> So you see, despite numerous meetings and brain storming back and forth, I still believe that all of this was completely off target … So in this case we did this formalistic play, which was good, but then we relaxed a bit. Thereafter, among the senior colleagues, we drew a pragmatic conclusion and went on. There was someone who quoted Shakespeare at the time: 'Much ado about nothing' or something like that … (10)

> …then some of us decided, within the department, to start a minor recurring training course … You see, it often comes down to quite strange results if we aren't part of the changing process … And when the colleagues 'over there' gained some knowledge about this matter, things definitely got better, at least from my point of view … Today this way of working is almost self-driven and I see it as a result completely independent of the investigation. (16)

## Time spent by the investigation team

In seven of 50 incident investigations, it was not possible to determine the amount of time spent by the team conducting the investigation. In 43 of 50 investigations, which had 217 recommendations, duration ranged from 12 to 150 man-hours. We grouped the different investigations by time spent on the investigation in order to examine if duration was a factor in implementation and at what level.

The investigations were of short duration (<40 h) (n=14), medium duration (41–80 h) (n=21) and long duration (>81 h) (n=8). We found that in the group with a short duration, actions had been taken on 25/55 recommendations, with 21/25 actions at the micro-level. In the group with a medium duration, actions had been taken on 43/116 recommendations with 28/43 actions at the micro-level, and in the long duration group, actions had been taken on 24/46 recommendations with 20/24 actions at the micro-level. The duration of the investigations differed by more than a factor 10 but this did not seem to influence actions taken to meet the recommendations, or at what level.

## DISCUSSION

This study has several strengths and limitations. The results presented show the advantages of using a design that combines content analysis with interviews to thereby achieve deeper understanding of the different aspects of the data. The semi-structured nature of the interviews seemed to encourage the respondents to elaborate and reflect freely on both questions and follow-up questions, which resulted in a substantial amount of qualitative data.

The coding scheme in the content analysis and the categories used in the data analysis could possibly result in a limited perspective of a more complex reality. It could be argued that an investigation is not complete before formal post-implementation follow-up has been carried out. However, we have not studied the effect of the implemented recommendations, since the focus of study was the gap between recommendation and implementation.

We do not draw general conclusions from this study. However, we expect our findings are not unique to the speciality (anaesthesiology) or type of hospital studied (university hospital), and thus believe that our findings may be valid for other Swedish hospitals, and possibly hospitals in countries with similar systems for investigating AEs.

This study shows that a clear majority of the recommendations presented to the CB were targeted at the micro-level of the organisation, even when the investigating team spent a considerable amount of time on their work. We suggest this finding reflects not that the micro-level is necessarily the most meaningful target of intervention but rather the investigating teams' understanding of how incidents happen. This is summarised in Hollnagel's two principles: WYLFIWYF ('What You Look For Is What You Find') and WYFIWYF ('What You Find Is What You Fix').[2 7 15] In this study, the causes the investigators sought are intimately linked to the linear causation model provided by the method available to them. The linear incident model inherent in the method provided by SALAR[5] and used in the investigations studied, identifies certain problems as relevant targets of intervention. This is not the first study to suggest that linear incident investigation methods tend to locate causes at the micro-level of the organisational hierarchy,[16] although we also see that what is found is not always fixed and it is not always the recommendations written in the reports that decide what will be fixed.

In the literature on healthcare system safety, much focus has been directed towards the sharp end, such as transition in care: change of shifts, change of ward and change in level of care.[17–19] Based on the findings from this study, we argue that in order to understand the successful implementation of recommendations following analyses of AEs, important factors found at the blunt end of the organisation should also be considered, such as changes in management positions or management continuity. Our results show that if the individual in a management position was the successor to the original CB, that individual had very little knowledge of an existing completed investigation, and understandably, took very little further action in addition to that taken by their predecessor.

Consequently, two principles—'close in space' and 'close in time'—seem to be important factors for closing the gap between recommendation and implementation when a model such as that employed in this university hospital, is used.

The finding that the event itself triggers organisational interventions regardless of the incident investigation recommendations requires further elaboration. This finding could be interpreted as lowering the organisational mandate of the analysis process, but it also complicates the process of conducting the analysis, especially if a model assuming a stable causal structure is employed. If organisational interventions are initiated simply as a result of the event, then the organisation essentially goes through a qualitative process of change as a result of that event. Consequently, this implies that the organisation is qualitatively different after the event than it was before.

Since the organisation did not record which recommendations had been implemented, our findings rely almost entirely on interviewee responses. This may introduce uncertainty about the reliability of the analysis results, but may also raise concern about how incident investigations are used by the organisation to improve patient safety. Based on the interviews, nearly half of all the recommendations had been implemented, regardless of how severe the organisation perceived the AE to be. A clear majority of these recommendations were at the micro-level, with the management position of the CB having very little effect.

The focus on success mechanisms also becomes a focus on system vulnerabilities and potential improvement. The finding suggesting that 'close in time' and 'close in space' actions are more likely to be implemented can indeed guide future work to improve the method of learning from AEs in the Swedish healthcare system. We suggest that future research and projects aimed at improving the quality of the system, focus on four aims. (1) Ways should be developed to institutionalise an organisational memory of AEs and the analyses following them so that the system becomes less sensitive to management continuity. (2) The target of analysis following AEs should be changed so as to spread suggested actions more evenly between the micro-, meso- and macro-levels of the organisation. This requires analysis focussing on interactions and relationships at higher organisational levels, and also investigation teams with basic competence in safety science and the interpretation of complex systems. (3) The gap between the investigation team and the investigated organisation should be closed. Based on the finding that other actions are taken in addition to those suggested by the incident investigation teams, we suggest future work is required on enhancing dialogue between analysis team and the organisation analysed. (4) Lessons should be learnt from incidents outside the formalised system. We suggest future research is conducted on the possible storytelling of past incidents in healthcare organisations. There may be many lessons that are never mentioned in formal investigations which can nevertheless be incorporated as part of organisational memory and everyday behaviour.

## CONCLUSIONS

This study seeks to understand the factors that lead to the successful implementation of recommendations suggested in incident investigations following AEs in a Swedish university hospital. Based on the findings, we conclude that continuity in management is an important factor for successful implementation of recommendations ('close in time'), as is a clear majority of the recommendations presented in the investigations being targeted at the micro-level of the organisation where the same applies to the recommendations that are actually implemented ('close in space'). The micro-level focus of the investigations is expected given the linear causal model underlying the method of analysis. For recommendations to be targeted towards the meso- and macro-levels of the organisation, the model used for investigation needs to seek causes at the level of organisational interactions and relationships. Furthermore, the AE itself triggers organisational interventions regardless of the recommendations made in the incident investigations. In addition, neither the time spent by the investigation team nor the position of the CB seems to contribute to the successful implementation of recommendations.

**Author affiliations**
[1]Department of Anesthesia and Intensive Care, Skåne University Hospital Lund, Lund, Sweden
[2]Department of Clinical Sciences, Lund University, Lund, Sweden
[3]Centre for Societal Resilience, Lund University, Lund, Sweden
[4]Centre for Risk Assessment and Management, Lund University, Lund, Sweden

**Acknowledgements** The authors would like to thank the staff at the office of the chief medical officer of the hospital for their contribution by providing all the necessary incident investigation reports.

**Contributors** JW developed the study, collected, analysed and interpreted the data, and wrote the manuscript. JB and PG contributed to study design, analysis of the data and revision of the manuscript.

**Funding** The project was partially funded by a grant from The Swedish National Patient Insurance Company.

**Competing interests** None.

**Ethics approval** The study was approved by the Regional Ethical Review Board in Lund, Sweden.

**Provenance and peer review** Not commissioned; externally peer reviewed.

**Data sharing statement** No additional data are available.

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
