## [Reviewer comments · BMJ Open]

This paper was submitted to the BMJQS but declined for publication following peer review. The authors addressed the reviewers' comments and submitted the revised paper to BMJ Open. The paper was subsequently accepted for publication at BMJ Open.

ARTICLE DETAILS

TITLE (PROVISIONAL)	Mind the gap between recommendation and implementation – principles and learnings in the aftermath of incident investigations. A semi-quantitative and qualitative study in search for factors that lead to successful implementation of recommendations.
AUTHORS	Wrigstad, Jonas; Bergström, Johan; Gustafson, Pelle

VERSION 1 - REVIEW

REVIEWER	Runciman, William Joanna Briggs Institute & The University of Adelaide, The Safety and Quality Research Unit
REVIEW RETURNED	02-Feb-2014

GENERAL COMMENTS	I think this paper is important because it provides information about systematic implementation of prevalent and corrective strategies - lacking or entirely haphazard in many facilities and jurisdictions. I found the germanic grammar somewhat distracting (I presume that that is what it is) and actually did a rough edit. I can mail through track changes to you if you wish. However, it could do with an edit to improve the colloquial English usage, remove "redundant qualifiers" and shorten the text.
--

- This manuscript received two reviews at the BMQS but the other referee had declined to make his comments public.

VERSION 1 – AUTHOR RESPONSE

Please let us know if you also find this version in need for further language review.